# Ceramides and Cardiovascular Risk Factors, Inflammatory Parameters and Left Ventricular Function in AMI Patients

**DOI:** 10.3390/biomedicines10020429

**Published:** 2022-02-11

**Authors:** Elena Michelucci, Silvia Rocchiccioli, Melania Gaggini, Rudina Ndreu, Sergio Berti, Cristina Vassalle

**Affiliations:** 1Institute of Clinical Physiology, National Research Council, Via G. Moruzzi 1, 56124 Pisa, Italy; emichelucci@ifc.cnr.it (E.M.); silvia.rocchiccioli@ifc.cnr.it (S.R.); mgaggini@ifc.cnr.it (M.G.); rudina.ndreu@ifc.cnr.it (R.N.); 2Fondazione CNR-Regione Toscana G Monasterio, 56124 Pisa, Italy; berti@ftgm.it

**Keywords:** ceramides, acute myocardial infarction, cardiovascular risk factors

## Abstract

**Background:** Ceramides, biologically active lipids correlated to oxidative stress and inflammation, have been associated with adverse outcomes in acute myocardial infarction (AMI). The purpose of this study was to assess the association between ceramides/ratios included in the CERT1 score and increased cardiovascular (CV) risk, inflammatory and left ventricular function parameters in AMI. **Methods:** high performance liquid chromatography-tandem mass spectrometry was used to identify Cer(d18:1/16:0), Cer(d18:1/18:0), and Cer(d18:1/24:1) levels and their ratios to Cer(d18:1/24:0), in 123 AMI patients (FTGM coronary unit, Massa, Italy). **Results:** Cer(d18:1/16:0): higher in female patients (<0.05), in patients with dyslipidemia (<0.05), and it directly and significantly correlated with aging, brain natriuretic peptide-BNP, erythrocyte sedimentation rate-ESR and fibrinogen. Cer(d18:1/18:0): higher in females (<0.01) and patients with dyslipidemia (<0.01), and increased according to the number of CV risk factors (considering hypertension, dyslipidemia and diabetes). Moreover, it significantly correlated with BNP, troponin at admission, ESR, C reactive protein-CRP, and fibrinogen. Cer(d18:1/24:1): significantly correlated with aging, BNP, fibrinogen and neutrophils. Cer(d18:1/16:0)/Cer(d18:1/24:0): higher in female patients (<0.05), and in patients with higher wall motion score index-WMSI (>1.7; ≤0.05), and in those with multivessel disease (<0.05). Moreover, it significantly correlated with aging, BNP, CRP, ESR, neutrophil-to-lymphocyte ratio-NRL, and fibrinogen. Cer(d18:1/18:0)/Cer(d18:1/24:0): higher in female patients (<0.001), and increased according to age. Moreover, it was higher in patients with lower left ventricular ejection fraction (<35%, ≤0.01), higher WMSI (>1.7, <0.05), and in those with multivessel disease (0.13 ± 0.06 vs. 0.10 ± 0.05 µM, <0.05), and correlates with BNP, ESR, CRP, fibrinogen and neutrophils, platelets, NLR, and troponin at admission. Multiple regression analysis showed that Cer(d18:1/16:0)/Cer(d18:1/24:0) and Cer(d18:1/18:0)/Cer(d18:1/24:0) remained as independent determinants for WMSI after multivariate adjustment (Std coeff 0.17, T-value 1.9, ≤0.05; 0.21, 2.6, <0.05, respectively). **Conclusion:** Distinct ceramide species are associated with CV risk, inflammation and disease severity in AMI. Thus, a detailed analysis of ceramides may help to better understand CV pathobiology and suggest these new biomarkers as possible risk predictors and pharmacological targets in AMI patients.

## 1. Introduction

Ceramides are a family of biologically active lipids, composed of sphingosine and a fatty acid, present in great amounts in cellular membranes [1].

The trend of lipidomic changes has been evidenced in general subject cohorts and in patients with cardiovascular disease in relation to aging and cardiometabolic risk, as well as their predictive value for adverse cardiovascular outcomes [2,3,4].

In particular, distinct ceramides resulted progressively higher in patients with stable angina, unstable angina, and acute myocardial infarction (AMI) with respect to healthy controls, and their levels appear related to disease severity (evaluated as the vessel stenosis percentage) [5,6]. 

Importantly, a lipidomic profile improves prediction of cardiovascular events upon traditional risk factors, as in the American ADVANCE trial and in the Italian Bruneck study [7,8]. 

These associations with clinical cardiovascular risk are supported by the multiple critical cellular pathways which may link ceramides to atherosclerosis, in particular those related to chronic inflammation and cytokines, oxidative stress, increased uptake of lipoproteins and accumulation of cholesterol within macrophages, and enhanced platelet activation [1,9]. Moreover, ceramides are among the main culprits for endothelial dysfunction by increasing oxidative stress and impairing nitric oxide release from endothelial cells [10]. It is also known that ceramide signaling affects cell growth, differentiation and apoptosis [9,11]. In particular, experimental data suggest that C2 ceramide (d18:1/2:0), a short-chain ceramide induces apoptosis in rat ventricular cardiomyocytes [12]. Conversely, a prolonged oxidative stress stimulus in cultured cardiomyocytes and ischemia of isolated hearts induces ceramide accumulation [13,14]. Interestingly, two imaging studies evidenced the localization and association of ceramides with the thin fibrotic plaques with a necrotic core, giving strength to the role of ceramides in plaque vulnerability and acute complications related to the rupture of inflammatory plaques [15,16].

However, despite the number of data suggesting the relevance of ceramides in cardiovascular pathophysiology, and their association with risk and severity of cardiovascular disease, the interpretation of ceramide results may be difficult for clinicians, as expressed by values of a number of ceramides, each one with its own reference values. Thus, the identification of few more significant ceramides may help to facilitate result interpretation and their adoption in future routine use in cardiovascular settings. Among all ceramide species, Cer18:1/C16:0, Cer18:1/C18:0 and Cer18:1/C24:1 were particularly reliable in terms of association with cardiovascular risk and prediction of adverse outcome in asymptomatic individuals and patients with cardiovascular disease [2,17,18].

Thus, some researchers proposed the calculation of a ceramide-based score (CERT1) that included these three species and their ratio to Cer18:1/C24:0 (a neutral specie used for normalization), efficaciously applied in a population of patients with acute and stable coronary artery disease for the prediction of mortality [19]. Moreover, the ratio calculation is not significantly influenced by an occasional variation in a single ceramide concentration, giving more strength to the results [20]. 

However, the relationship of the components of the CERT1 score with cardiovascular risk factors and left ventricular function in AMI patients remains to be further determined.

Thus, the main aim of this study was to examine whether there were significant associations between CERT1 score components and cardiovascular risk factors, in particular inflammatory-related parameters, and left ventricular function parameters in patients with AMI.

## 2. Materials and Methods

### 2.1. Population Characteristics and Data Acquisition

The overall studied population included 123 ST-segment elevation myocardial infarction (STEMI) enrolled at the Ospedale del Cuore G. Pasquinucci-Clinical Cardiology Department (Massa, Italy). STEMI was defined according to SC/ACCF/AHA/WHF guidelines for STEMI criteria and management [21]. Patients underwent coronary angiography with subsequent percutaneous coronary intervention within 90 min from admission to the intensive care unit. Information on each patient (e.g., demographic data, cardiovascular history, cardiovascular risk factors and laboratory parameters) were extracted by the Hospital-computerized database. Hypertension was defined as follows: blood pressure > 140/90 mmHg or current use of antihypertensive drugs; dyslipidemia as: use of lipid-lowering treatments or fasting low-density lipoprotein levels > 150 mg/dL. Type 2 diabetes (T2D) was defined when use of antidiabetic treatment, fasting glucose > 126 mg/dL (7 mmol/L) on two separate tests before the acute event, or finding of HbA1c > 6.49%. Smoking history was ascertained by the clinical anamnesis as a current and former smoking habit. Left ventricular (LV) function was estimated by ejection fraction (EF) calculation through echocardiographic 2-D measurement (modified Simpson’s rule with biplane planimetry), and is defined as the ratio of the difference between the end-systolic and the end-diastolic volumes and the end-diastolic volume itself. It represents a dimensionless parameter, expressed as a percentage (%). For the wall motion score index (WMSI), to each myocardial segment is assigned a score from 1 to 4 (from normokinesia to hypokinesia, akinesia, dyskinesia) by using a standard transthoracic echocardiography sequence. Then, the WMSI is calculated by dividing the sum of the segmental values by the number of myocardial segments (*n* = 16). A WMSI of 1.0 is considered normokinetic.

Patients were considered eligible to be enrolled in the study based on the following inclusion criteria: (1) male and female adults; (2) patients admitted to the Hospital coronary care unit for chest pain and subsequently proven STEMI; (3) signed informed consent obtained from each patient (or from their relatives if the patient was unable to sign it). Exclusion criteria were: (1) systemic diseases (e.g., inflammatory, cancer); (2) lack of a written Informed Consent; (3) patients without available HbA1c data. Standard therapy, including administration of aspirin, beta-blockers, ACE-inhibitors, diuretics, and statins, was given to eligible patients. The study was approved by the local ethics committee (number 19214, 11 February 2021).

### 2.2. Plasma Processing

All chemicals, standards and stock solutions are described in Appendix A. For ceramides identification we analyzed one hundred twenty-three plasma samples, stored at −80 °C, that were thawed at room temperature and immediately subjected to lipid extraction and analysis. 10 μL of sample were put in a 1.5 mL microcentrifuge tube and 200 μL of 0.003 μM *N*,*N*-dimethylsphingosine (d18:1) (DMS), selected as internal standard (ISTD), in cold MeOH (4 °C) were added. The white precipitate thus obtained was centrifuged at 13,000 rpm for 20 min at 4 °C in a Microcentrifuge Heraeus Biofuge Fresco (Thermo Scientific, Waltham, MA, USA) and the supernatant was withdrawn and transferred into a glass vial for the subsequent high performance liquid chromatography-tandem mass spectrometry (HPLC-MS/MS) analysis (ISTD concentration 0.00285 μM in vial).

### 2.3. HPLC-MS/MS Analysis

HPLC-MS/MS analyses of plasma samples and external standard calibration curve (see Appendix A) were performed using a Nexera X2 HPLC system (Shimadzu, Kyoto, Japan) combined with a QTrap 5500 mass spectrometer (SCIEX, Concord, ON, Canada) equipped with a Turbo V ESI source. Vials were put in a refrigerated autosampler at 5 °C and each calibration level/plasma extract was injected in duplicate (injection volume 0.5 μL). To prevent carry-over, two needle wash solutions were used: MeOH/i-PrOH 50/50 and MeOH/CHCl_3_ 1/2. Gradient elution at a flow rate of 0.2 mL/min was performed on a Kinetex column packed with a C8 phase (100 mm × 2.1 mm, 1.7 μm, 100 Å) equipped with a guard column packed with C8 phase (3 mm ID), both Phenomenex (Torrance, CA, USA). Mobile phase A was MeOH/H_2_O/i-PrOH 50/45/5 and phase B was MeOH/i-PrOH 50/50, both with 5 mM ammonium formate. The gradient elution program was: 0 min, 65% B; 0.5 min, 65% B; 10 min, 100% B; 15 min, 100% B; 15.1 min, 65% B; 20 min, 65% B. The column temperature was 45 °C. The mass spectrometer was set in positive ion mode and the operation conditions were as follows: curtain gas 30 psi, ion spray voltage 5 kV, probe temperature 200 °C, ion source gas 1 and gas 2 30 psi, declustering potential 100 V, entrance potential 10 V, collision cell exit potential 19 V, collisional gas N_2_. A set up by direct infusion [22] was carried out in order to optimize the collision energies (CEns) for each STD/Cer species and to choose the relative product ions to be used in the MRM/SRM (multiple reaction monitoring/selected reaction monitoring) analysis. A dwell time of 20 msec was used for each transition Q1/Q3 (see Appendix A). Analyst Software 1.6.3 (SCIEX, Concord, ON, Canada) was used to collect data.

Absolute concentrations of Cer lipid species were calculated considering their area ratio (lipid peak area/ISTD peak area) and interpolating it within the calibration curve of the corresponding external standard (equation y = 12.063x + 0.0273, with R^2^ = 0.9994). MultiQuant 2.1 software (SCIEX, Concord, ON, Canada) was used for lipid quantitation (results in Appendix A).

### 2.4. Statistical Analysis 

Continuous variables were reported as mean ± SD, categorical variables as a number (percentage). Owing to skewness, Log transformations were used for some parameters to perform statistical analyses. Then, log-transformed values were back-transformed for data presentation. Comparisons between continuous variables were evaluated by Student’s t-test, while differences between categorical variables were performed by using the Chi-square analysis. Comparisons among variables of three or more independent groups were analyzed using the ANOVA, and Scheffè test used for post hoc analysis. Regression analysis was performed to assess the relationship between continuous variables. Multiple regression analysis was also applied to verify the effect of significant variables in determining echocardiographic biomarkers of left ventricular dysfunction (EF and WMSI). A *p*-value < 0.05 was chosen as the level of significance.

## 3. Results

### 3.1. Patient Characteristics

A total of 123 patients with AMI admitted to the Heart Hospital Pasquinucci of Fondazione Gabriele Monasterio (Massa, Italy) were enrolled in the study. Demographic, clinical characteristics and laboratory parameters were reported in Table 1. Mean age corresponded to 69 ± 12 years; as expected there was a prevalence of male patients (*n* = 87, 71%) and of multi- against mono-vessel disease (59 vs. 41%).

### 3.2. Ceramide Distribution and Levels in AMI

Nine ceramide species were assessed in the studied population (Figure 1), from which the species included in the CERT1 score were further evaluated for subsequent analysis.

The distribution curves of ceramides in AMI were analyzed according to Gaussian distributions, and skewness and kurtosis for each ceramide species was also reported in Figure 2. 

Plasma levels of ceramides in AMI patients resulted higher respect to those found in a control subjects (healthy subjects, *n* = 15, 5 males, comparable for age, samples ran in parallel to AMI specimens), as follow: Cer(d18:1/16:0) corresponding to 0.9 ± 0.3 vs. 0.7 ± 0.2 µM, in AMI patients and controls, respectively *p* < 0.05; Cer(d18:1/18:0) 0.3 ± 0.2 vs. 0.15 ± 0.1 µM, *p* < 0.01; Cer(d18:1/24:1) 1.4 ± 0.6 vs. 0.9 ± 0.4 µM, *p* < 0.01; their ratios Cer(d18:1/18:0)/Cer(d18:1/24:0) 0.12 ± 0.06 vs. 0.06 ± 0.03 µM, *p* ≤ 0.001, and Cer(d18:1/24:1)/Cer(d18:1/24:0) 0.5 ± 0.2 vs. 0.3 ± 0.07 µM, *p* < 0.01, but not Cer(d18:1/16:0)/Cer(d18:1/24:0) (Figure 3). 

### 3.3. Cer(d18:1/16:0)

Cer(d18:1/16:0) resulted higher in female patients (1 ± 0.4 vs. 0.9 ± 0.2 µM in males, *p* < 0.05; Figure 4), in patients with dyslipidemia (1 ± 0.3 vs. 0.9 ± 0.3 µM in males, *p* < 0.05), and increased with aging (Table 2). Moreover, it directly and significantly correlated with brain natriuretic peptide-BNP, erythrocyte sedimentation rate-ESR and fibrinogen (Table 2).

### 3.4. Cer(d18:1/18:0)

Cer(d18:1/18:0) was higher in females (0.4 ± 0.3 vs. 0.3 ± 0.1 µM in males, *p* < 0.01; Figure 3) and in patients with dyslipidemia (0.4 ± 0.2 vs. 0.3 ± 0.1 µM vs. no-dislipidemia, *p* < 0.01), and increased according to the number of CV risk factors (considering hypertension, dyslipidemia and T2D) (0.2 ± 0.1, 0.3 ± 0.1, 0.4 ± 0.2 µM, for 0, 1 and >1 risk factors, respectively, *p* ≤ 0.01; Figure 5). Moreover, it significantly correlated with BNP, troponin at admission, ESR, C reactive protein-CRP, and fibrinogen (Table 2).

### 3.5. Cer(d18:1/24:1) 

Cer(d18:1/24:1) significantly correlated with aging, BNP, fibrinogen and neutrophils (Table 2).

### 3.6. Cer(d18:1/16:0)/Cer(d18:1/24:0) (CERT-R1)

Cer(d18:1/16:0)/Cer(d18:1/24:0) resulted higher in female patients (0.38 ± 0.13 vs. 0.32 ± 0.12 µM in males, *p* < 0.05; Figure 4), and in patients with a higher WMSI (>1.7; 0.4 ± 0.21 vs. 0.33 ± 0.11 µM, *p* ≤ 0.05), and in those with multivessel disease compared with multivessel disease patients (0.13 ± 0.06 vs. 0.1 ± 0.05 µM, *p* < 0.05) (Figure 6 and Figure 7, respectively). Moreover, it increased according to age (Table 2) and it significantly correlated with BNP, CRP, ESR, neutrophil-to-lymphocyte ratio-NRL, and fibrinogen (Table 2).

### 3.7. Cer(d18:1/18:0)/Cer(d18:1/24:0) (CERT-R2)

Cer(d18:1/18:0)/Cer(d18:1/24:0) was higher in female patients (0.14 ± 0.07 vs. 0.1 ± 0.05 µM in males, *p* < 0.001; Figure 4), and increased according to age. Moreover, it was higher in patients with lower EF (<35%) (0.16 ± 0.07 vs. 0.11 ± 0.06 µM, *p* ≤ 0.01), higher WMSI (>1.7; 0.15 ± 0.07 vs. 0.11 ± 0.06 µM, *p* < 0.05), and in those with multivessel disease (0.13 ± 0.06 vs. 0.10 ± 0.05 µM, *p* < 0.05) (Figure 6, Figure 7 and Figure 8 respectively), and correlates with BNP, ESR, CRP, fibrinogen and neutrophils, platelets, NLR, and troponin at admission (Table 2).

### 3.8. Cer(d18:1/18:0)/Cer(d18:1/24:0 (CERT-R3)

Cer(d18:1/24:1)/Cer(d18:1/24:0) resulted higher in female patients (0.55 ± 0.07 vs. 0.47 ± 0.05 µM in males, *p* < 0.05; Figure 4), and in patients with higher WMSI (>1.7; 0.58 ± 0.25 vs. 0.49 ± 0.16 µM, *p* ≤ 0.05), and in those with multivessel disease (0.53 ± 0.18 vs. 0.47 ± 0.16 µM, *p* ≤ 0.05) (Figure 6 and Figure 7, respectively), and significantly increased according to age, BNP, ESR, CRP, fibrinogen, neutrophils, NRL, and troponin at admission (Table 2).

### 3.9. Ceramides as Deteminants of Echocardiographic Left Ventricular Dysfunction in AMI Patients

Multiple regression analysis showed that CERT-R1 (Std coeff 0.17, *T*-value 1.9, *p* ≤ 0.05) and CERT-R2 (0.21, 2.6, *p* < 0.05, respectively) remained as independent determinants for WMSI after multivariate adjustment (parameters included in the univariate analysis were: age, gender, diabetes, hypertension, dyslipidemia, BMI, creatinine, fibrinogen, monocytes, platelets, lymphocytes, neutrophils, ESR, NLR, CRP, troponin I-at admission, and ceramides and their ratio) (Table 3).

## 4. Discussion

It has been postulated that ceramides promote aggregation and absorption of lipoproteins (especially LDL) in the arterial wall, which renders the lipid-rich core larger and more prone to rupture, a fact confirmed by two recent imaging studies, which involve ceramides in the plaque rupture [15,16,23]. Interestingly, in a previous recent study, higher level of plasma Cer(d18:1/24:0) has been found to be associated with a greater severity of left anterior descending artery stenosis in a mixed group of acute and stable CAD patients, confirming the possible association of ceramides with coronary disease severity [6]. Our study adds confirmation to these findings, evidencing for the first time a correlation of ceramides with multivessel disease (an index of disease severity), EF (index of left ventricular contractility), WMSI (an index of wall motion) and BNP (index of wall strain and dysfunction) in an AMI cohort. This relationship is particularly important in view of the link between ceramides and left ventricular dysfunction and heart failure [24]. In particular, some data indicate that ceramides accumulate in the failing myocardium, and increased concentration is relievable in the bloodstream. Conversely, inhibition of de novo ceramide synthesis reduces cardiac remodelling, and improvement in myocardial systolic function. In vitro findings suggested that the changes in ceramide are related to hypoxia, oxidative stress and inflammatory processes. All together, these results suggested that de novo ceramide synthesis is closely related to cardiac remodelling and dysfunction, also indicating that inhibition of ceramide synthesis may represent an intriguing possibility in the treatment and prevention of the failing heart [25]. Moreover, the fact that all ceramides considered and their ratios were associated to BNP, together with the finding that Cer(d18:1/18:0)/Cer(d18:1/24:0) and Cer(d18:1/24:1)/Cer(d18:1/24:0) were associated with elevated hs-TnT, suggested the possible role of ceramides in both myocardial necrosis and dysfunction following AMI. 

Specific ceramides and/or their ratios were also related to specific clinical characteristics and also to inflammatory and cardiac biomarkers. Previous experimental and clinical data suggested that distinct ceramides may be elevated with hypertension, type 2 diabetes, and insulin resistance [26,27]. In this context, we observed that Cer(d18:1/18:0) increased with the number of major cardiovascular risk factors. Moreover, as expected, patients with dyslipidemia presented higher ceramide levels. In fact, ceramides increased in the presence of high levels of total and LDL cholesterol, as they promote LDL accumulation and aggregation at the wall vessel level [23]. However, ceramides may retain even better predictivity of residual risk over traditional lipids, thus identifying high-risk patients also among those with low LDL-C, as evidenced in coronary artery disease patients [4,19]. This fact is important, as being ceramides modifiable by diet and exercise as well as by drugs (e.g., small molecule inhibitors of key enzymes, anti-inflammatory agents), it is therefore plausible that ceramide evaluation in the clinical setting may be relevant, especially if antioxidants/anti-inflammatory trials will give positive results [28,29,30]. Possible underlying mechanisms by which specific plasma ceramides might contribute to the pathophysiology of atherosclerosis are not fully cleared, although many data suggest that ceramide circulating changes might be a reflex of cardiovascular inflammation. Accordingly, we found a significant association between ceramides and different inflammatory-related biomarkers (see Table 2). Different clinical and experimental studies have found positive correlations between plasma ceramide levels and inflammatory biomarkers and cytokines, suggesting the interconnection between inflammation and ceramide production [4,31]. Moreover, it is known that S-SMase activity (the enzyme which catalyzes the cleavage of sphingomyelinase to ceramides) is mediated by inflammation [32]. In particular, multiple experimental and clinical findings have suggested an association between ceramides and C reactive protein [4,33,34]. Specifically, ceramides were previously found to be significantly correlated with C-reactive protein in the SPUM-ACS population, in agreement with our findings [19]. Moreover, according to different experimental findings, different ceramide species appeared to be involved in a variety of key signaling pathways related to atherogenesis, which reflect the alterations related to lipid profile, inflammation, myocardial necrosis, and myocardial dysfunction, as confirmed by the correlation of biomarkers related to these biological processes with ceramides that we observed in our population [35,36,37].

To note, the relationship between distinct ceramides and risk factors showed specificity according to the acyl chain length and saturation. In particular, very long-chain (e.g., (d18:1/24:0)) species appear more beneficial than long-chain species (e.g., (d18:1/16:0) and (d18:1/18:0)). Accordingly, long-chain ceramides appeared pro-apoptotic, whereas very long-chain ceramides seems to exert anti-apoptotic effect in an experimental model (Caenorhabditis elegans) [38]. In addition, Cer(d18:1/24:1) behaved less adversely with respect to Cer(d18:1/24:0) [18]. Thus, these results underline the importance of evaluating each individual specie of ceramides to establish the specific risk profiles which may occur during CV disease onset and progression. Nonetheless, the identification of a panel of ceramides are more significant in terms of risk profile and prognostic role, as those included in the CERT1 score may be useful in terms of translational power to the clinical setting, and are easier to interpret for clinicians [1,20].

Regarding gender-related differences, it is not still clear whether there are differences in ceramides between males and females, nor a shared consensus on the optimal cutoffs and reference ranges of ceramides are to be used. In our population, we observed higher ceramide levels in women. However, it is important to consider that the totality of our female patients were in the postmenopausal status, thus likely reflecting estradiol lack as an inducer of ceramide unbalance. In fact, a very recent study evidenced an increase of Cer(d18:1/24:0) and Cer(d18:1/24:1) with aging, especially in postmenopausal women [39,40]. Moreover, estradiol appears to inversely correlated with Cer(d18:1/24:1) in women, and the incubation with estradiol (10 nM, 24 h) in cancer cells expressing estrogen receptors decreased ceramide accumulation, which further supported the hypothesis of ceramide modulation by estradiol [3]. 

To note, all patients enrolled were Caucasians; however, as it is possible that some circulating ceramide values differed across races/ethnicities [41], the present results may not be generalizable to other racial/ethnic groups. 

## 5. Conclusions

The present study supported the relationship between selected ceramide species/ratios in the CERT1 score with the number of CV risk factors and left ventricular function in AMI patients, with the involvement of inflammatory-related biomarkers and pathways. Specifically, Cer(d18:1/16:0) was higher in female patients, in patients with dyslipidemia, and it directly and significantly correlated with aging, BNP, ESR and fibrinogen. Cer(d18:1/18:0) was higher in females and patients with dyslipidemia, and increased according to the number of CV risk factors (considering hypertension, dyslipidemia and diabetes). Moreover, it significantly correlated with BNP, troponin at admission, ESR, CRP, and fibrinogen. Cer(d18:1/24:1) was significantly correlated with aging, BNP, fibrinogen and neutrophils. Concerning the ratios, Cer(d18:1/16:0)/Cer(d18:1/24:0) was higher in female patients, and in patients with higher WMSI (>1.7), and in those with multivessel disease. Moreover, it significantly correlated with aging, BNP, CRP, ESR, NRL, and fibrinogen. Cer(d18:1/18:0)/Cer(d18:1/24:0) was higher in female patients, and increased according to age. Moreover, it was higher in patients with lower left ventricular ejection fraction (<35%), higher WMSI (>1.7), and in those with multivessel disease, and correlated with BNP, ESR, CRP, fibrinogen and neutrophils, platelets, NLR, and troponin at admission. Multiple regression analysis showed that Cer(d18:1/16:0)/Cer(d18:1/24:0) and Cer(d18:1/18:0)/Cer(d18:1/24:0) remained independent determinants for WMSI after multivariate adjustment for significant variables at the univariate analysis. 

In conclusion, the present findings suggest that the assessment of serum ceramides might improve the risk-stratification of patients with STEMI, making easier the development of a personalized approach. In particular, the documentation of the new associations between plasma ceramide levels and coronary disease severity and left ventricular dysfunction in AMI patients may suggest how these lipids are to be further studied as potential targets to improve cardiac function, and thus clinical outcomes in patients with acute coronary heart disease, also in view of their capacity to be modified by life-style (e.g., diet and exercise) as well as pharmacological tools [28]. Actually, their measurements in the clinical practise is not so diffuse in view of the still expensive techniques, requirement of skilled technicians, low availability of instruments in all routine laboratories and lack of standardization, beyond the difficulties involved in understanding results that are complex for clinicians. In this context, the use of few reliable ceramides (as those in the CERT1 score) may facilitate measurement, interpretation, and translational passage of these biomarkers in the clinical practice.

## Figures and Tables

**Figure 1 biomedicines-10-00429-f001:**
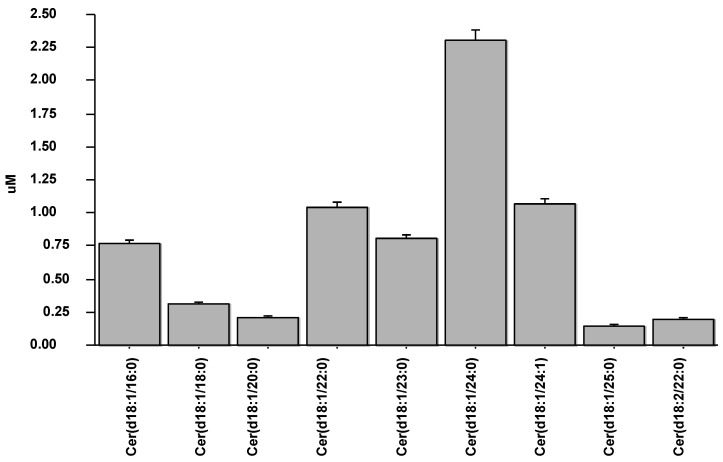
Values of nine ceramide species measured in the studied population.

**Figure 2 biomedicines-10-00429-f002:**
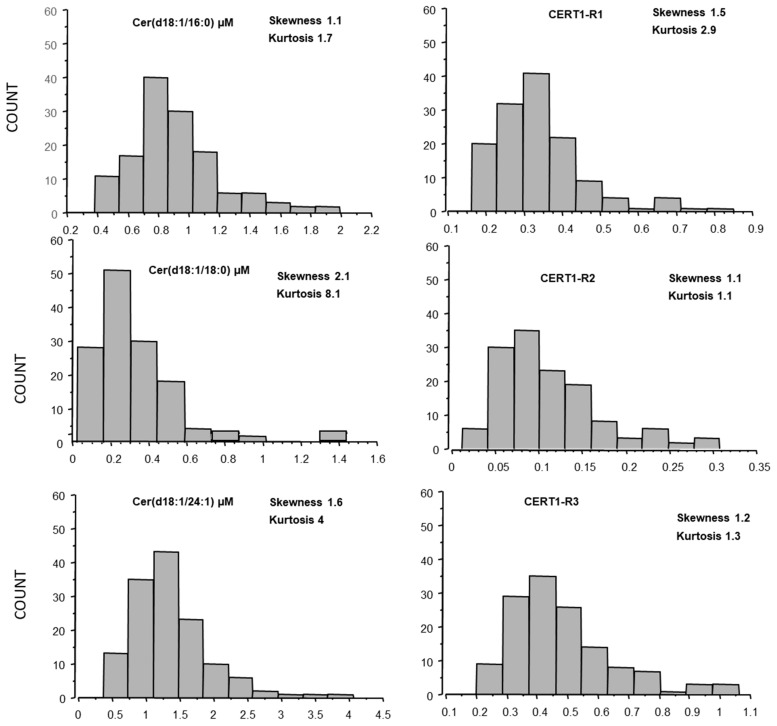
Ceramide distribution (skewness and kurtosis) in AMI patients.

**Figure 3 biomedicines-10-00429-f003:**
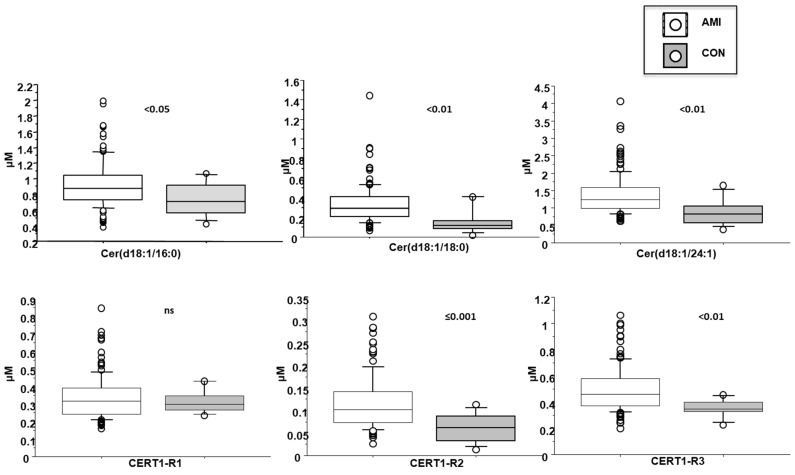
Ceramides in AMI patients vs. controls. Median, interquartile, outliers and extremes of ceramides are given.

**Figure 4 biomedicines-10-00429-f004:**
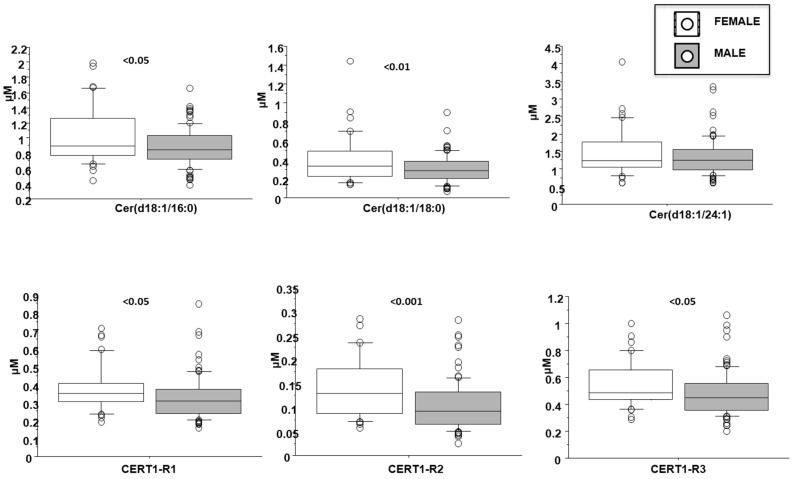
Ceramides according to gender. Median, interquartile, outliers and extremes of ceramides are given.

**Figure 5 biomedicines-10-00429-f005:**
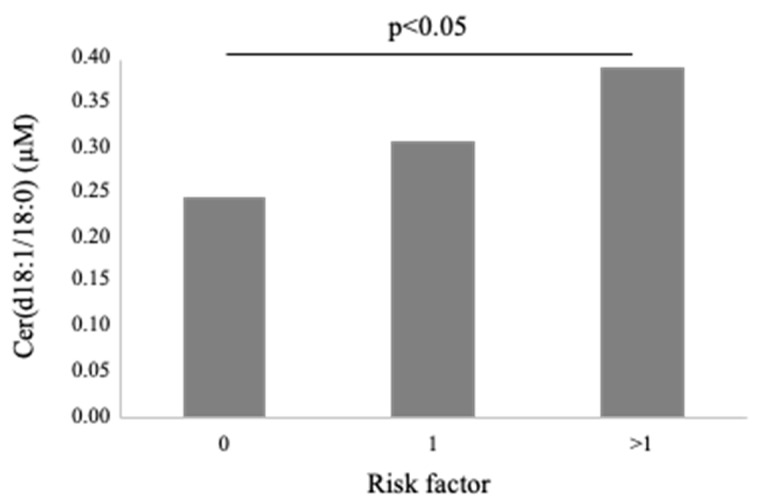
Cer(d18:1/18:0) according to number of traditional cardiovascular risk factors (hypertension, dyslipidemia and T2D).

**Figure 6 biomedicines-10-00429-f006:**
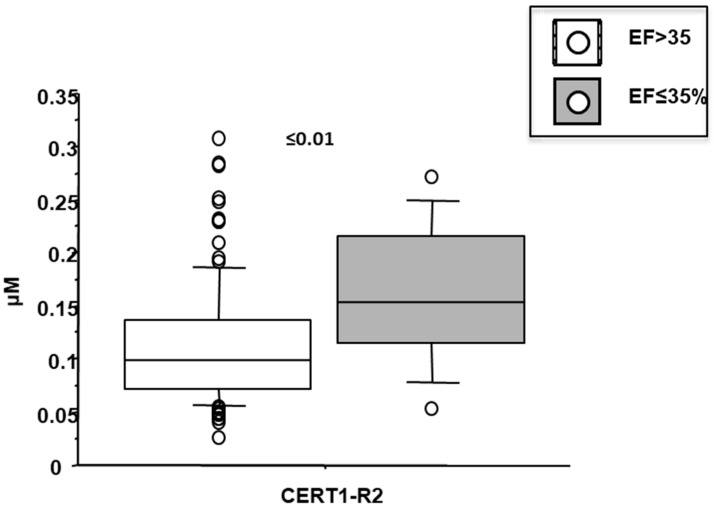
Ceramide ratios (R2) according to left ventricular ejection fraction (EF). Median, interquartile, outliers and extremes of ceramides are given.

**Figure 7 biomedicines-10-00429-f007:**
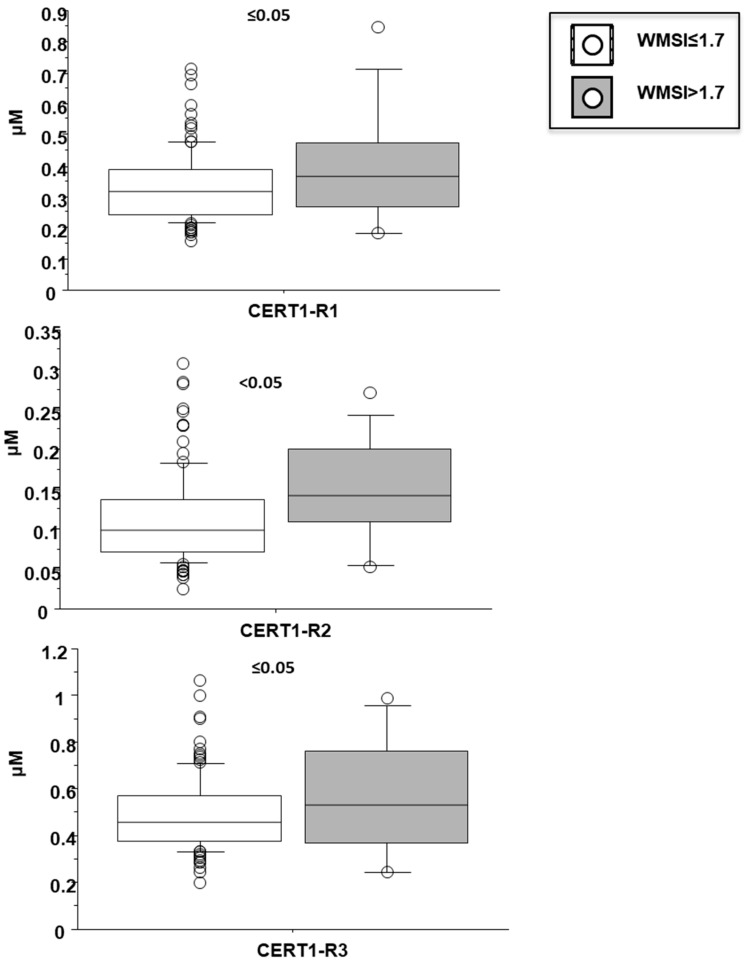
Ceramide ratios according to wall motion score index (WMSI). Median, interquartile, outliers and extremes of ceramides are given.

**Figure 8 biomedicines-10-00429-f008:**
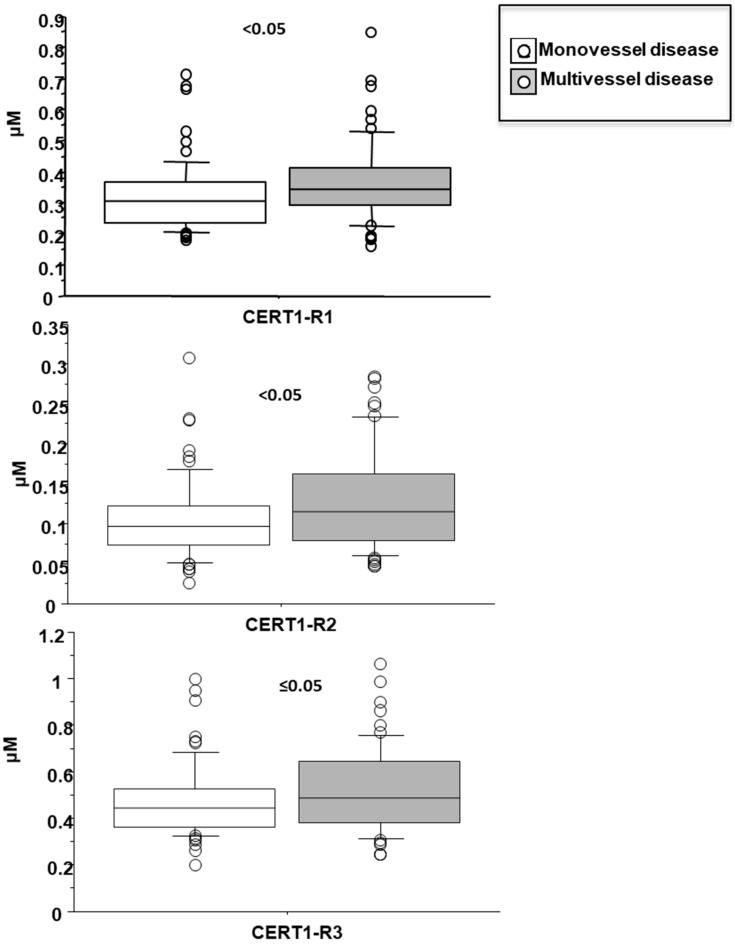
Ceramide ratios according to mono- vs. multi-vessel disease. Median, interquartile, outliers and extremes of ceramides are given.

**Table 1 biomedicines-10-00429-t001:** Demographic, clinical and laboratory parameters in the 123 AMI patients.

Males		87 (71)
Age (years)		69 ± 12
Body mass index (kg/m^2^)		27 ± 5
CV risk factors		
	Type 2 Diabetes	22 (18)
	Hypertension	76 (62)
	Dyslipidemia	61 (49)
	Current/ex smoking habit	56 (45)
	Ejection fraction (%)	50 ± 9
	Wall Motion Score Index	1.2 ± 0.23
	Obesity	21 (17)
Laboratory parameters		
	Creatinine (mg/dL)	1 ± 0.8
	Brain Natriuretic Peptide (ng/L)	217 ± 319
	Fibrinogen (mg/dL)	376 ± 122
	Monocytes (10^9^/L)	0.9 ± 0.7
	Platelets (10^9^/L)	231 ± 75
	Lymphocytes (10^9^/L)	2.1 ± 1.9
	Neutrophils (10^9^/L)	6.5 ± 2.6
	Erythrocyte Sedimentation Rate (mm/h)	27 ± 26
	Neutrophil-to-lymphocyte-ratio	4.0 ± 2.8
	C Reactive Protein (mg/dL)	3.4 ± 4
	Troponin I (at admission) (μg/L)	26 ± 53
Multivessel disease		59 (48)

Data are expressed as mean ± SD or number (%).

**Table 2 biomedicines-10-00429-t002:** Correlation between distinct ceramide species and ratios and CV risk factors, inflammatory parameters and left ventricular dysfunction in AMI patients.

	Cer(d18:1/16:0)	Cer(d18:1/18:0)	Cer(d18:1/24:1)	Cer(d18:1/16:0)/Cer(d18:1/24:0)	Cer(d18:1/18:0)/Cer(d18:1/24:0)	Cer(d18:1/24:1)/ Cer(d18:1/24:0)
Age (years)	r = 0.22 *p* < 0.05	ns	r = 0.17 *p* ≤ 0.05	r = 0.50 *p* < 0.001	r = 0.29 *p* < 0.01	r = 0.49 *p* < 0.001
Ejection fraction (%)	ns	ns	ns	ns	r = 0.17 *p* ≤ 0.05	ns
Wall Motion Score Index	ns	ns	ns	r = 0.22 *p* < 0.05	r = 0.27 *p* < 0.01	r = 0.19 *p* < 0.05
Brain Natriuretic Peptide (ng/L)	r = 0.25 *p* < 0.01	r = 0.28 *p* < 0.01	r = 0.29 *p* < 0.01	r = 0.45 *p* < 0.001	r = 0.45 *p* < 0.001	r = 0.55 *p* < 0.001
C Reactive Protein (mg/dL)	ns	r = 0.37 *p* < 0.01	ns	r = 0.31 *p* < 0.05	r = 0.50 *p* < 0.001	r = 0.30 *p* < 0.05
Erythrocyte Sedimentation Rate (mm/h)	r = 0.20 *p* < 0.05	r = 0.23 *p* < 0.05	ns	r = 0.40 *p* < 0.001	r = 0.41 *p* < 0.001	r = 0.37 *p* < 0.001
Fibrinogen (mg/dL)	r = 0.28 *p* < 0.01	r = 0.41 *p* < 0.001	r = 0.25 *p* < 0.01	r = 0.25 *p* < 0.01	r = 0.47 *p* < 0.001	r = 0.31 *p* < 0.001
Neutrophils (10^9^/L)	ns	ns	r = 0.19 *p* ≤ 0.05	ns	r = 0.20 *p* < 0.05	r = 0.24 *p* < 0.05
Platelets (10^9^/L)	ns	ns	ns	ns	r = 0.18 *p* ≤ 0.05	ns
Neutrophil-to- lymphocyte-ratio	ns	ns	ns	r = 0.20 *p* < 0.05	r = 0.18 *p* ≤ 0.05	r = 0.24 *p* < 0.05
Troponin I (at admission) (μg/L)	ns	r = 0.24 *p* < 0.05	ns	ns	r = 0.34 *p* < 0.001	r = 0.19 *p* < 0.05

**Table 3 biomedicines-10-00429-t003:** Multiple regression between WMSI levels and univariate significant variables.

Variable	STD Coefficient	T Value	*p*	STD Coefficient	T Value	*p*
CERT-R1				0.17	1.9	≤0.05
CERT-R2	0.21	2.26	<0.05			
T2D	0.27	2.9	<0.05	0.29	3.1	<0.01
Platelet count	0.13	1.5	ns	0.16	1.9	ns

## Data Availability

Data are available on request from the authors.

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
