# Peer review of "Ceramides and Cardiovascular Risk Factors, Inflammatory Parameters and Left Ventricular Function in AMI Patients"

_biomedicines, 2022, doi:10.3390/biomedicines10020429_

Round 1

Reviewer 1 Report

The authors describe their work on the association between ceramides and increased cardiovascular risk, inflammatory and left ventricular function parameters in AMI. It was found that specific ceramides and ratios correlated with different cardiometabolic risk factors, inflammatory-related parameters, disease severity (multi- versus mono-vessel disease) and brain natriuretic peptide, a biomarker of myocardial stress and adverse prognosis. The authors concluded that distinct ceramide species are associated with CV risk, inflammation and disease severity in AMI. Thus, a detailed analysis of ceramides may help to better understand CV pathobiology and suggest these new biomarkers as possible risk predictors and pharmacological targets in AMI patients. This is an interesting and straightforward study. Appropriate methodology has been employed and the conclusions appear to be justified based on the data at hand. The manuscript is written well. I have a few recommendations for consideration.

  1. Please provide a clear hypothesis to be tested in the study.
  2. The authors should elaborate and emphasize the novelty aspect of their work as well as the clinical applicability of their findings.
  3. Is there any role in aging in influencing ceramide levels in at risk patients?
  4. The influence of ethnicity in the association between ceramide levels and CV risk could be discussed.

Author Response

The authors describe their work on the association between ceramides and increased cardiovascular risk, inflammatory and left ventricular function parameters in AMI. It was found that specific ceramides and ratios correlated with different cardiometabolic risk factors, inflammatory-related parameters, disease severity (multi- versus mono-vessel disease) and brain natriuretic peptide, a biomarker of myocardial stress and adverse prognosis. The authors concluded that distinct ceramide species are associated with CV risk, inflammation and disease severity in AMI. Thus, a detailed analysis of ceramides may help to better understand CV pathobiology and suggest these new biomarkers as possible risk predictors and pharmacological targets in AMI patients. This is an interesting and straightforward study. Appropriate methodology has been employed and the conclusions appear to be justified based on the data at hand. The manuscript is written well. I have a few recommendations for consideration.

    1.  

Please provide a clear hypothesis to be tested in the study.

The introduction seciton was modified according to reviewer suggesitons

    1.  

The authors should elaborate and emphasize the novelty aspect of their work as well as the clinical applicability of their findings.

Conclusion secton was modified according to reviewer suggestions

    1.  

Is there any role in aging in influencing ceramide levels in at risk patients?

No, no significant role of aging for this relationship was observed.

    1.  

The influence of ethnicity in the association between ceramide levels and CV risk could be discussed.

This point was added in the discussion section (with one more reference)

Reviewer 2 Report

The present study reports an important translational approach in cardiovascular impact-related dysfunction. The main observations include ceramides composition in their saturated/unsaturated/Carbon length in fatty acid/sphingosine in ceramide, inflammatory factors and oxidative stress factor in acute myocardial infarction (AMI). The authors have checked the association between ceramides and cardiovascular (CV) risk, inflammatory and left ventricular function parameters in AMI. Cer(d18:1/16:0), Cer(d18:1/18:0), and Cer(d18:1/24:1) were analyzed with their ratios to Cer(d18:1/24:0), in clinical AMI patients. Results showed specific ceramides and ratios correlated with cardiovascular risk factors, inflammatory-related parameters, disease severity (multi/mono-vessel disease) and brain natriuretic peptide, myocardial stress marker and prognosis. Specific ceramide species are associated with CV risk, inflammation and disease severity in AMI. Ceramides are some related factors for CV disease.

As discussed,  the current observation is very limited to only Cer(d18:1/16:0), Cer(d18:1/18:0), and Cer(d18:1/24:1) . However, previous literatures show a variety of caramide species are partially related to CVD or neurodegeneration. Although the present limits in the data availability, the study can be further encourage to enhance the manuscript with additional evidences. 

Major issue: More detailed species of ceramides should analyzed and added.

Author Response

The present study reports an important translational approach in cardiovascular impact-related dysfunction. The main observations include ceramides composition in their saturated/unsaturated/Carbon length in fatty acid/sphingosine in ceramide, inflammatory factors and oxidative stress factor in acute myocardial infarction (AMI). The authors have checked the association between ceramides and cardiovascular (CV) risk, inflammatory and left ventricular function parameters in AMI. Cer(d18:1/16:0), Cer(d18:1/18:0), and Cer(d18:1/24:1) were analyzed with their ratios to Cer(d18:1/24:0), in clinical AMI patients. Results showed specific ceramides and ratios correlated with cardiovascular risk factors, inflammatory-related parameters, disease severity (multi/mono-vessel disease) and brain natriuretic peptide, myocardial stress marker and prognosis. Specific ceramide species are associated with CV risk, inflammation and disease severity in AMI. Ceramides are some related factors for CV disease.

As discussed,  the current observation is very limited to only Cer(d18:1/16:0), Cer(d18:1/18:0), and Cer(d18:1/24:1). However, previous literatures show a variety of caramide species are partially related to CVD or neurodegeneration. Although the present limits in the data availability, the study can be further encourage to enhance the manuscript with additional evidences. 

Major issue: More detailed species of ceramides should analyzed and added.

We wanted to thank the reviewer for this important observaton. However, in this specific manuscript we wanted to focus on ceramides included in the CERT1 score, also in view of the translational capacity by the use of a simple score. In fact, the identification of a restricted number of few relevant ceramides species and/or their ratio may facilitate comprehension of results by medical professionals and the translational passage in the clinical practice as we explained in the introduction section as follow:

More studied are Cer(d18:1/16:0), Cer(d18:1/18:0) and Cer(d18:1/24:1), and their ratios to Cer(d18:1/24:0), recently utilized to calculate the CERT1 score, proposed and efficaciously applied in a population of patients with acute and stable coronary artery disease for the prediction of mortality [10]. Normally the interpretation of ceramide results is difficult, as expressed by values of a number of ceramides, each one with its own reference values. Thus, the proposal of a SCORE aimed to render easier the interpretation of results, especially in the clinical setting [11]. Moreover, the ratio calculation is not significantly influenced by an occasional variation in a single ceramide concentration, giving more strength to results [11].”

Nonetheless, other ceramides have been measured in this population, and these parameters will be included in a future study, more focused on pathophysiological roles of ceramide species.

Reviewer 3 Report

Michelucci et al. presented a manuscript about potential correlations between ceramides, inflammation, and oxidative stress in patients with AMI. The general idea of ​​the work is good, but unfortunately, it needs some key improvements, such as below:
1. In the Abstract, the Results part does not contain the most important observations that have been recorded in the work. This part is in need of improvement.
2. The first part of the Introduction requires some correction. It does not clearly show the key role of ceramides in CVD. So it requires a deeper knowledge and a more accurate description of the relevance of the research performed.
3. Materials and methods part, the description of the population lacks the abbreviation PCI.
4. In part the Results
- description for table 1: "mean +/- DS" (should be mean +/- SD);
- Figure 2 - it is not known, which graph is panel A and which panel B. Please specify the captions;
- The data shown in Figures 2 and 4 are difficult to read. I recommend changing the appearance of the figure.
5. In Conclusions - the existence of a correlation was concluded, but it is necessary to clarify what is the relationship between the examined parameters.

Author Response

Michelucci et al. presented a manuscript about potential correlations between ceramides, inflammation, and oxidative stress in patients with AMI. The general idea of ​​the work is good, but unfortunately, it needs some key improvements, such as below:

1. In the Abstract, the Results part does not contain the most important observations that have been recorded in the work. This part is in need of improvement.

Rewritten according to reviewer suggestions

2. The first part of the Introduction requires some correction. It does not clearly show the key role of ceramides in CVD. So it requires a deeper knowledge and a more accurate description of the relevance of the research performed.

The introduction has been improved with addition of more references, following reviewer suggestions

3. Materials and methods part, the description of the population lacks the abbreviation PCI.

Checked and specified

4. In part the Results

- description for table 1: "mean +/- DS" (should be mean +/- SD);

Checked and corrected

- Figure 2 - it is not known, which graph is panel A and which panel B. Please specify the captions;
The data shown in Figures 2 and 4 are difficult to read. I recommend changing the appearance of the figure.

According to reviewer suggestions figures 2 and 4 were splitted in figure 2 and 3, and 5, 6 and 7, respectively

5. In Conclusions - the existence of a correlation was concluded, but it is necessary to clarify what is the relationship between the examined parameters.

Conclusion section was rewritten, emphasizing the novelty aspect of the study, the clinical applicability of findings, and reporting the specific relationship between the examined parameters.

Round 2

Reviewer 2 Report

The authors answered without additional experimental data.

This is not fair for further recommendation.

Still impact of the present study is needed at least.

Author Response

The authors answered without additional experimental data.

This is not fair for further recommendation.

Still impact of the present study is needed at least.

Response to reviewer: In this revised version we added new data (new figure 1) showing values of nine species of ceramides measured in the studied population.

Moreover, a new sentence in the conclusion section was added to underline the potential impact of ceramide evaluation in the clinical practise